# Generative Recorrupted-to-Recorrupted: An Unsupervised Image Denoising Network for Arbitrary Noise Distribution

Yukun Liu [1,†], Bowen Wan [1,†], Daming Shi [1,*] and Xiaochun Cheng [2]

1  College of Computer Science and Software Engineering, Shenzhen University, Shenzhen 518060, China
2  Computer Science Department, Middlesex University, Hendon, London NW4 4BT, UK
*  Correspondence: dshi@szu.edu.cn
†  These authors contributed equally to this work.

**Abstract:** With the great breakthrough of supervised learning in the field of denoising, more and more works focus on end-to-end learning to train denoisers. In practice, however, it can be very challenging to obtain labels in support of this approach. The premise of this method is effective is that there is certain data support, but in practice, it is particularly difficult to obtain labels in the training data. Several unsupervised denoisers have emerged in recent years; however, to ensure their effectiveness, the noise model must be determined in advance, which limits the practical use of unsupervised denoising.n addition, obtaining inaccurate noise prior to noise estimation algorithms leads to low denoising accuracy. Therefore, we design a more practical denoiser that requires neither clean images as training labels nor noise model assumptions. Our method also needs the support of the noise model; the difference is that the model is generated by a residual image and a random mask during the network training process, and the input and target of the network are generated from a single noisy image and the noise model. At the same time, an unsupervised module and a pseudo supervised module are trained. The extensive experiments demonstrate the effectiveness of our framework and even surpass the accuracy of supervised denoising.

**Keywords:** image denoising network; unsupervised; pseudo supervised

## 1. Introduction

Image denoising is a traditional topic in the field of image processing and is an essential basis for success in other vision tasks. Noise is an obvious cause of image interference, and an image may have a wide variety of noise in practice, which significantly degrades the image quality. Previous denoising methods can be separated into three main types according to different inputs and training methods: supervised training denoising, self-supervised training denoising and unsupervised training denoising [1–4]. A noisy image can be represented by $y = x + n$. Our task is to design a denoiser to remove the noise from the image.

The denoising convolutional neural network (DNCNN) [1] is a benchmark of the use of deep learning for image denoising. It also introduced residual learning and batch normalization, which speed up the training process and boost the denoising performance. The fast and flexible denoising neural network (FFDNET) [5] treats the noisy model as a prior probability distribution, such that it can effectively handle a wide range of noise levels. The convolutional blind denoising (CBDNET) [2] went further than the (FFDNET) [5], aimed at real photographs, though synthesized and real images were both used in the training.

A common treatment for the above methods is that it all needs to take noisy–clean image pairs during the training. However, in some scenarios such as medical and biological imaging, there often few clean images, leading to an infeasibility of the above methods. To this end, the noise-to-noise (N2N) method [6] was the first to reveal that deep neural

networks (DNNs) can be trained with pairs of noisy–noisy images instead of noisy–clean images. In other words, training can be conducted with only two noisy images that are captured independently in the same scene. The N2N can be used in many tasks [7–11], because it creatively addressed the dependency on clean images. Unfortunately, pairs of corrupted images are still difficult to obtain in dynamic scenes with deformations and image quality variations.

To further bring the N2N into practice, some research [3,12–14] concluded that it is still possible to train the network without using clean images if the noise between each region of the image is independent. The Neighbor2Neighbor (NBR2NBR) method [14] proposed a new sampling scheme to achieve better denoising effects with a single noisy image. The advantage of this approach is that it does not need a prior noise model, such as the Recorrupted-to-Recorrupted (R2R) [15], nor does it lose image information, like the Noise2Void (N2V) self-supervised method N2V [3].

Nevertheless, [3,12–15] are valid under the assumption that the noise on each pixel is independent from each other, which means that they are not as effective in dealing with noise in real scenes as supervised denoising. To tackle more complex noise, some unsupervised methods were proposed. Noise2Grad (N2G) [4] extracted noise by exploiting the similar properties of noise gradients and noisy image gradients, and then adding the noise to unpaired clean images to form paired training data. In [16], a new type of unsupervised denoising through optimal transport theory was constructed. It is worth noting that although [4,16] no longer subject the end-to-end learning approach to pairing clean–noisy images, they still need to collect many clean images.

In order to solve the above problems, a new denoising mode that achieves unsupervised denoising without requiring noise prior and any clean images is proposed. In one epoch of our training network, we obtain a residual image containing noise through the difference between the network input and output and use a random mask image to reduce the influence of natural image information in the residual image. The generative noise model can be obtained by the above operation. In the second step, we put the model into the pseudo supervised module and the recorrupted-to-recorrupted module to train the same network. Eventually, after several iterations of network training, we will obtain a more realistic noise model and a more accurate denoiser. More details of our proposed generative recorrupted-to-recorrupted framework can be found in Figure 1.

The remainder of this paper is organized as follows. In Section 2, we introduce the related work. Then, the details of our method are given in Section 3, followed by the experiments in Section 4. In Section 5, we discuss the results, with the conclusions drawn in Section 6.

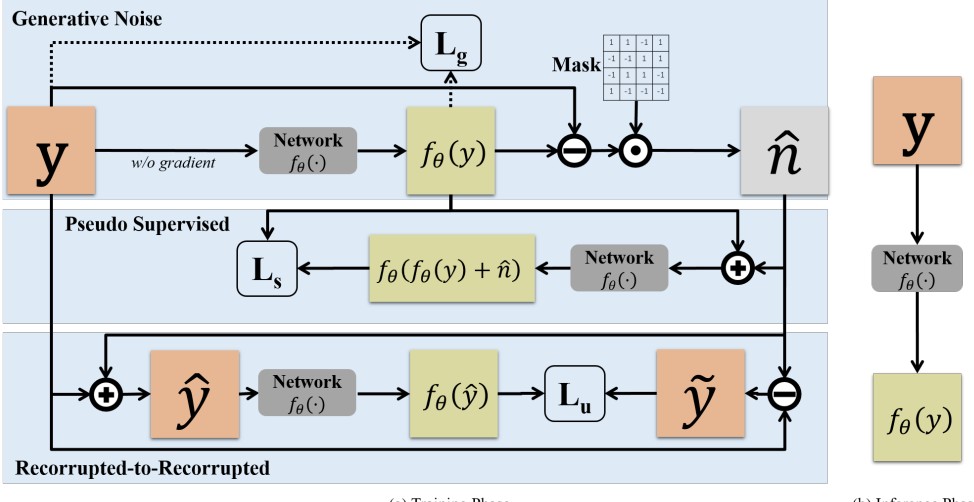

(a) Training Phase          (b) Inference Phase

**Figure 1.** The framework of generative recorrupted-to-recorrupted (GR2R). (**a**) Overall training process. $y$ is the observed noisy image, the generative noise $\hat{n}$ in pseudo supervised (PS) and Recorrupted2Recorrupted (R2R) is obtained by element-wise multiplication of random mask $m$ and residual map $y - f_\theta(y)$. The neural networks in the three modules update the same parameter $\theta$ in one network. Moreover, the regular loss $L_g$ is used to stabilize the training phase, the supervised loss $L_s$ avoids randomly generated noise from affecting unsupervised denoising and $L_u$ represents unsupervised loss. (**b**) Inference using the trained denoising model. The denoising network generates denoised images directly from the noisy images $y$ of the test set without additional operations.

## 2. Problem Statement

### 2.1. Supervised Training

With the rapid development of deep learning, many supervised learning methods are applied to image denoising. The DNCNN [1] successfully applied a 17-layer deep neural network to image denoising tasks by introducing residual learning. After that, a series of more efficient and complex neural networks succeeded in denoising tasks. Unlike the DNCNN [1], the FFDNET [5] was more efficient in denoising, and the CBDNET [2] can handle more complex real noise. Without considering constraints, the above supervised denoising methods can be expressed by the following equation:

$$\underset{\theta}{\arg\min}\, L(f_\theta(y), x). \tag{1}$$

where the $y$, $x$, $f$ and $L$ are noisy images, clean images, denoising model and loss function, respectively.

However, these methods all require clean images as the target of training the neural network, and then optimize the parameter $\theta$ by calculating the gap between the network output and the target, so as to obtain a better denoising model.

The N2N [6] revealed that the noisy/true image pairs used to train the DNN can be replaced by noisy/noisy image pairs. The corrupted pairs are represented by $y$ and $z$, where $n_1$ and $n_2$ are uncorrelated. There are two main principles for the N2N to successfully train a network with a paired noisy image: the first is that the optimal solution obtained by network training is a mean value solution; the second is that the mean value of most noise is close to zero. Because the N2N only requires that the ground truth possesses some "clean" statistical values, and does not require that every ground truth be "clean", the optimal

solution of this loss function is obtained as the arithmetic mean of the measured values (expectation) of the measurements:

$$\underset{\theta}{\text{argmin}} \mathbb{E}_{z,y} \{L(f_\theta(y), z)\}$$
$$y = x + n_1$$
$$z = x + n_2.$$

(2)

Although the N2N alleviated the dependence on clean images, pairs of noisy images are still difficult to obtain.

### 2.2. Self-Supervised Training

To eliminate the N2N [6] restrictions in dynamic scene denoising, some methods utilized the original features of noisy images to construct auxiliary tasks to make self-supervision as effective as standard supervision.

The N2V [3] offered a blind-spot network which can denoise using only a noisy image, and its main principle is to use the correlation among pixels to predict missing pixels. However, when using this method, some pixel information is lost and the denoising effect is not ideal. The Self2Self [17] can realize image denoising by dropout in the case of only one noisy image. Unlike the N2V, it used lost information as targets, but its training time was extremely long. A training model was required to test each image, which limits its practical application. The NBR2NBR [14] was the most recent self-supervised denoising method, which can be viewed as an advanced version of the N2N [6] via a novel image sampling scheme to get rid of the requirement of paired noisy images. The following equation shows the reason why denoising is effective without clean targets under the condition that the noise mean is zero:

$$\mathbb{E}_{x,y}\|f_\theta(y) - x\|_2^2 = \mathbb{E}_{a,y}\|f_\theta(y) - a\|_2^2 + const$$
$$+ Cov((f_\theta(y) - x), (a - x)).$$

(3)

In the N2N [6], $a = z$, because $n_1$ and $n_2$ are independent covariance terms that will disappear. Similarly in NBR2NBR [14], $a$ and $y$ are replaced by two adjacent noisy sub-images, where covariance terms disappear under the local correlation of pixels and the assumption of uncorrelated noise. Moreover, in N2V [3], it is denoised by a blind spot network, which removes the covariance term by assuming that the noise is uncorrelated on adjacent pixels.

### 2.3. Unsupervised Training

A category of unsupervised denoising requires unpaired clean–noisy images to train the denoiser. The N2G [4] obtained an approximate noise model by measuring the distance between the noise gradient generated after denoising and the original noisy image gradient and then added it to the unpaired clean images for training. However, the approach requires clean images and significant approximation time for the real noise distribution. Another approach [16] utilized the WGAN-gp [18] to measure the distance between the denoised image output by the generator and the unpaired clean image to optimize the denoiser.

Another category of unsupervised denoising is when certain assumptions about the noisy model are required. Moreover, the AmbientGAN [19], a method of training generative adversarial networks, also eliminated the reliance on clean images, using a measurement function to generate a noisy image, which was then fed into the discriminator for comparison with the original noisy image. The Noiser2Noise (Nr2N) [20] re-destroyed the original noisy images through the noise prior as the input $\hat{y} = y + \hat{n}$ to the network, and the original noisy images $y$ were used as training targets. Then, the R2R [15] overcame the disadvantage of the N2N [6] and generated paired noisy images by introducing a prior noise model. In particular, for a single noisy image, it assembles noisy pairs using known

noise levels. Although the R2R is close to the N2N [6] in the denoising effect, it is limited in applications with real noisy images because the level and type of noise from real noisy images are difficult to measure.

## 3. Theoretical Framework

Our framework consists of three modules: Recorrupted2Recorrupted (R2R), generative noise (GN), and pseudo supervised (PS). These three modules share a neural network parameter during training, which means they are trained simultaneously but in a different order in each network training iteration. First, we generate a simulated noise through GN. The second step involves adding noise to PS and R2R and updating the network parameters at the same time. In the following iteration, GN will produce more realistic noise by using the newly updated parameters.

### 3.1. Recorrupted2Recorrupted Module

R2R [15] is an unsupervised denoising method which uses a training scheme that does not require noisy image pairs or clean target images. The approach destroys a single noisy image through the noise prior to form the paired noisy images required in N2N [6]. Given the noisy observation $y$ and noise prior $n$, R2R aims to minimize the following empirical risk:

$$\begin{aligned} &\operatorname*{argmin}_{\theta} L(f_\theta(\hat{y}), \tilde{y}) \\ &\hat{y} = y + n' \\ &\tilde{y} = y - n'. \end{aligned} \tag{4}$$

where $f_\theta(\cdot)$ denotes the denoising model with parameter $\theta$. This method assumes that noise $n'$ is determined before training network; however, it is difficult to accurately estimate the noise model in real noisy image denoising, which affects the performance of the denoiser.

The loss of Equation (4) is closely related to the equation used in supervised learning:

$$\operatorname*{argmin}_{\theta} L(f_\theta(\hat{y}), x). \tag{5}$$

Proofs are as follows:

$$\begin{aligned} \mathbb{E}_{n,n'}\left\{\|f_\theta(\hat{y}) - \tilde{y}\|_2^2\right\} &= \mathbb{E}_{n,n'}\left\{\|f_\theta(\hat{y}) - x - n + n'\|_2^2\right\} \\ &= \mathbb{E}_{n,n'}\left\{\|f_\theta(\hat{y}) - x\|_2^2\right\} - A + B \\ &= \mathbb{E}_{n,n'}\left\{\|f_\theta(\hat{y}) - x\|_2^2\right\} \end{aligned}$$

$$A = 2\mathbb{E}_{n,n'}\left\{(n - n')^\top (f_\theta(\hat{y}) - x)\right\} \tag{6}$$

$$B = \mathbb{E}_{n,n'}\left\{(n - n')^\top (n - n')\right\} \tag{7}$$

where $n$, $n'$, $x$ are the noise in the observed image, the noise prior and the clean image, respectively. In Equation (6), as long as $n$ and $n'$ are independent, it can be inferred that $n + n'$ in $f_\theta(\hat{y})$ and $n - n'$ are independent. According to the assumption of N2N [6] that the noise mean is zero, it can be found that Equation (6) is equal to 0. In Equation (7), because the variance in the noise prior $n'$ is equal to the variance in the noise $n$, Equation (7) is exactly equal to zero.

Next, going a step further, a generative noise method is used to implement unsupervised denoising without the noise prior.

### 3.2. Generative Noise Module

A natural idea is to generate the noisy model at training time. First, we generate a residual image by following Equation $y - f_\theta(y)$, which contains real noise. However, because the residual image also contains a large amount of image feature information, it is

difficult to accurately model noise. So, in the second step, we introduce a random mask map $m$, which is a vector of the same size and dimension as the noisy image, and it obeys the following distribution:

$$m = \begin{cases} 1 & p = 0.5 \\ -1 & p = 0.5. \end{cases} \tag{8}$$

Then, the generated noise during training can be represented as follows:

$$\hat{n} = (y - f_\theta(y)) \odot m. \tag{9}$$

where $\odot$ represents element-wise multiplication. It can be seen from Equation (8) that $m$ is randomly generated and has a mean value of zero. From Equation (9), it can be deduced that the mean value of $\hat{n}$ is zero. Even if the original real noise $n$ does not satisfy the assumption that the noise from [6] has a mean of zero, the generated noise can be forced to satisfy the assumption of zero mean during network training iterations by Equation (9). Additionally, the operation may produce a slight error in estimating the real noise model, so the pseudo supervised module is designed to reduce this error.

The input and target of our Recorrupted2Recorrupted are represented by the following equation:

$$\begin{aligned} \hat{y} &= y + \hat{n} \\ \tilde{y} &= y - \hat{n}. \end{aligned} \tag{10}$$

This shows that the generated $\hat{n}$ and $n$ are irrelevant, so $n + \hat{n}$ and $n - \hat{n}$ are unrelated. At this time, under the generated noise $\hat{n}$ scheme, Equation (6) vanishes as in Recorrupted2Recorrupted. Because the variance of $m$ is one, Equation (7) can be transformed into the following equation:

$$\mathbb{E}_{n,\hat{n}} \left\{ \|f_\theta(y) - y\|_2^2 \right\} + const. \tag{11}$$

Therefore, an optimal denoising criterion to achieve the same effect of supervised denoising can be expressed as

$$\operatorname*{argmin}_{\theta} L(f_\theta(\hat{y}), \tilde{y}) + \operatorname*{argmin}_{\theta} L(f_\theta(y), y). \tag{12}$$

### 3.3. Pseudo Supervised

Note that both Equation (4) and Equation (12) are equivalent to supervised denoising Equation (5). However, in Equation (5), the extra noise introduced by $\hat{y}$ will affect the accuracy in the denoiser, so R2R [15] adopts the Monte Carlo approximation to solve the above problem. However, the averaging of multiple forward processes in R2R will not only greatly reduce the denoising speed on the test set but also affect the denoising accuracy. In addition, the noise we generate will have a certain error, so we design a supervised-like approach to address the effect of generative noise. Specifically, we use the generated $f_\theta(y)$ in GN as the "clean" target and $f_\theta(y) + \hat{n}$ as the input to train the denoiser. During training, because GN stops the update of $\theta$, $f_\theta(y)$ gradually approaches the clean image without affecting the stability of training. The pseudo supervised loss as follows:

$$\operatorname*{argmin}_{\theta} L(f_\theta(f_\theta(y) + \hat{n}), f_\theta(y)). \tag{13}$$

The total loss function of our method can be expressed as

$$\begin{aligned} L &= L_u + L_g + \gamma \cdot L_s \\ &= \|f_\theta(\hat{y}) - \tilde{y}\|_2^2 + \|f_\theta(y) - y\|_2^2 + \gamma \cdot \|f_\theta(f_\theta(y) + \hat{n}) - f_\theta(y)\|_2^2 \end{aligned} \tag{14}$$

where $f_\theta$ is a denoising network with arbitrary network design, and $\gamma$ is a hyperparameter controlling the strength of the pseudo supervised term.

## 4. Experiments

In this section, we evaluate our GR2R framework, with significant improvement to the denoising quality of previous work.

**Training Details.** We use the same modified U-Net [21] architecture as [6,14,22]. The batch size is 10. We use Adam [23] as our optimizer. The initial learning rate is 0.0003 for synthetic denoising in sRGB space and 0.0001 for real-world denoising. All models are trained on a server using Python 3.8.5, Pytorch 1.6 and Nvidia Tesla K80 GPUs.

**Datasets for Synthetic Denoising.** Following the setting in [6,14], we select 44,328 images with sizes between $256 \times 256$ and $512 \times 512$ pixels from the ILSVRC2012 [24] validation set as the training set. To obtain reliable average PSNRs, we also repeat the test sets Kodak [25], BSD300 [26] and Set14 [27] by 10, 3 and 20 times, respectively. Thus, all methods are evaluated with 240, 300 and 280 individual synthetic noise images. Specifically, we consider four types of noise in sRGB space: (1) Gaussian noise with a fixed noise level $\sigma = 25$ , (2) Gaussian noise with a variable noise level $\sigma \in [5, 50]$ , (3) Poisson noise with a fixed noise level $\lambda = 30$, (4) Poisson noise with a variable noise level $\lambda \in [5, 50]$.

**Datasets for Real-World Denoising.** Following the setting in [14], we take the Smartphone Image Denoising Dataset (SIDD) [28] collected by five smartphone cameras in 10 scenes under different lighting conditions for real-world denoising in raw-RGB space, which has about 30,000 noisy images and 200 scene instances, of which 160 scene instances are used as training set and 40 scene instances are used as test set. We use only raw-RGB images in SIDD Medium Dataset for training and use SIDD Validation and Benchmark Datasets for validation and testing.

**Details of Experiments.** For the baseline, we consider two supervised denoising methods (N2C [21] and N2N [6]). Both of these methods require paired input. Additionally, we compare our proposed GR2R with a traditional approach (BM3D [29]) and eight self-supervised or unsupervised denoising algorithms (Self2Self [17], Noise2Void (N2V) [3], Laine19 [30], Noisier2Noise [20], DBSN [31], R2R [15], NBR2NBR [14] and B2UB [22]), all of which require only a single noisy image as input. The difference is that both Laine19 and R2R require a noise prior.

### 4.1. Results for Synthetic Denoising

The quantitative comparison results of synthetic denoising for Gaussian and Poisson can be seen in Table 1. Whether the Gaussian and Poisson noise level is fixed or variable, our approach significantly outperforms the traditional denoising method BM3D and most self-supervised denoising methods on the BSD300 dataset, even beyond the supervised learning methods of paired input. On the other two small test sets (KODAK and SET14), our method is also close to Laine19-pme [30], which is a method that requires the same explicit noise modeling as the R2R [15]. Our method is to iteratively keep learning the generative noise closer to the real data rather than some single distribution of noise. Therefore, for Gaussian or Poisson noise, our denoising effect is slightly insufficient. In addition, compared to Laine19, we use a mask to train the noise model, which has the potential to ignore the central pixel, resulting in only a few output pixels that can contribute to the loss function. However, explicit noise modeling means a strong prior, leading to a poor performance on real data. The following experiments on real-world datasets also illustrate this problem.

**Table 1.** Quantitative denoising results on synthetic datasets in sRGB space. The highest PSNR(dB)/SSIM among unsupervised denoising methods is highlighted in bold.

| Noise Type | Method | Input | KODAK | BSD300 | SET14 |
|---|---|---|---|---|---|
| Gaussian $\sigma = 25$ | Baseline, N2C [21] | Paired Input | 32.43/0.884 | 31.05/0.879 | 31.40/0.869 |
| | Baseline, N2N [6] | Paired Input | 32.41/0.884 | 31.04/0.878 | 31.37/0.868 |
| | BM3D [29] | Non Noise Prior | 31.87/0.868 | 30.48/0.861 | 30.88/0.854 |
| | Self2Self [17] | Non Noise Prior | 31.28/0.864 | 29.86/0.849 | 30.08/0.839 |
| | N2V [3] | Non Noise Prior | 30.32/0.821 | 29.34/0.824 | 28.84/0.802 |
| | Laine19-mu [30] | Noise Prior | 30.62/0.840 | 28.62/0.803 | 29.93/0.830 |
| | Laine19-pme [30] | Noise Prior | **32.40/0.883** | 30.99/0.877 | **31.36/0.866** |
| | Noisier2Noise [20] | Noise Prior | 30.70/0.845 | 29.32/0.833 | 29.64/0.832 |
| | DBSN [31] | Noise Independent | 31.64/0.856 | 29.80/0.839 | 30.63/0.846 |
| | R2R [15] | Noise Prior | 32.25/0.880 | 30.91/0.872 | 31.32/0.865 |
| | Ours | Non Noise Prior | 32.34/0.882 | **31.08/0.879** | 31.20/0.862 |
| Gaussian $\sigma \in [5, 50]$ | Baseline, N2C [21] | Paired Input | 32.51/0.875 | 31.07/0.866 | 31.41/0.863 |
| | Baseline, N2N [6] | Paired Input | 32.50/0.875 | 31.07/0.866 | 31.39/0.863 |
| | BM3D [29] | Non Noise Prior | 32.02/0.860 | 30.56/0.847 | 30.94/0.849 |
| | Self2Self [17] | Non Noise Prior | 31.37/0.860 | 29.87/0.841 | 29.97/0.849 |
| | N2V [3] | Non Noise Prior | 30.44/0.806 | 29.31/0.801 | 29.01/0.792 |
| | Laine19-mu [30] | Noise Prior | 30.52/0.833 | 28.43/0.794 | 29.71/0.822 |
| | Laine19-pme [30] | Noise Prior | 32.40/0.870 | 30.95/0.861 | **31.21/0.855** |
| | DBSN [31] | Noise Independent | 30.38/0.826 | 28.34/0.788 | 29.49/0.814 |
| | R2R [15] | Noise Prior | 31.50/0.850 | 30.56/0.855 | 30.84/0.850 |
| | Ours | Non Noise Prior | **32.46/0.875** | **31.13/0.867** | 31.02/0.856 |
| Poisson $\lambda = 30$ | Baseline, N2C [21] | Paired Input | 31.78/0.876 | 30.36/0.868 | 30.57/0.858 |
| | Baseline, N2N [6] | Paired Input | 31.77/0.876 | 30.35/0.868 | 30.56/0.857 |
| | BM3D [29] | Non Noise Prior | 30.53/0.856 | 29.18/0.842 | 29.44/0.837 |
| | Self2Self [17] | Non Noise Prior | 30.31/0.857 | 28.93/0.840 | 28.84/0.839 |
| | N2V [3] | Non Noise Prior | 28.90/0.788 | 28.46/0.798 | 27.73/0.774 |
| | Laine19-mu [30] | Noise Prior | 30.19/0.833 | 28.25/0.794 | 29.35/0.820 |
| | Laine19-pme [30] | Noise Prior | **31.67/0.874** | **30.25/0.866** | **30.47/0.855** |
| | DBSN [31] | Noise Independent | 30.07/0.827 | 28.19/0.790 | 29.16/0.814 |
| | R2R [15] | Noise Prior | 30.50/0.801 | 29.47/0.811 | 29.53/0.801 |
| | Ours | Non Noise Prior | 30.69/0.855 | 29.73/0.856 | 29.23/0.831 |
| Poisson $\lambda \in [5, 50]$ | Baseline, N2C [21] | Paired Input | 31.19/0.861 | 29.79/0.848 | 30.02/0.842 |
| | Baseline, N2N [6] | Paired Input | 31.18/0.861 | 29.78/0.848 | 30.02/0.842 |
| | BM3D [29] | Non Noise Prior | 29.40/0.836 | 28.22/0.815 | 28.51/0.817 |
| | Self2Self [17] | Non Noise Prior | 29.06/0.834 | 28.15/0.817 | 28.83/0.841 |
| | N2V [3] | Non Noise Prior | 28.78/0.758 | 27.92/0.766 | 27.43/0.745 |
| | Laine19-mu [30] | Noise Prior | 29.76/0.820 | 27.89/0.778 | **28.94/0.808** |
| | Laine19-pme [30] | Noise Prior | 29.60/0.811 | 27.81/0.771 | 28.72/0.800 |
| | DBSN [31] | Noise Independent | 29.60/0.811 | 27.81/0.771 | 28.72/0.800 |
| | R2R [15] | Noise Prior | 29.14/0.732 | 28.68/0.771 | 28.77/0.765 |
| | Ours | Non Noise Prior | **30.19/0.839** | **29.26/0.839** | 28.90/0.822 |

In addition, Figure 2 shows that our method retains more natural image features while denoising. Specifically, the N2C, N2N and N2V are visually smoother than the clean images, while the NBR2NBR and B2UB are more detailed with our method of denoising, i.e., natural images are more detailed. However, our method works better on oversized images, providing more details than the NBR2NBR and B2UB.

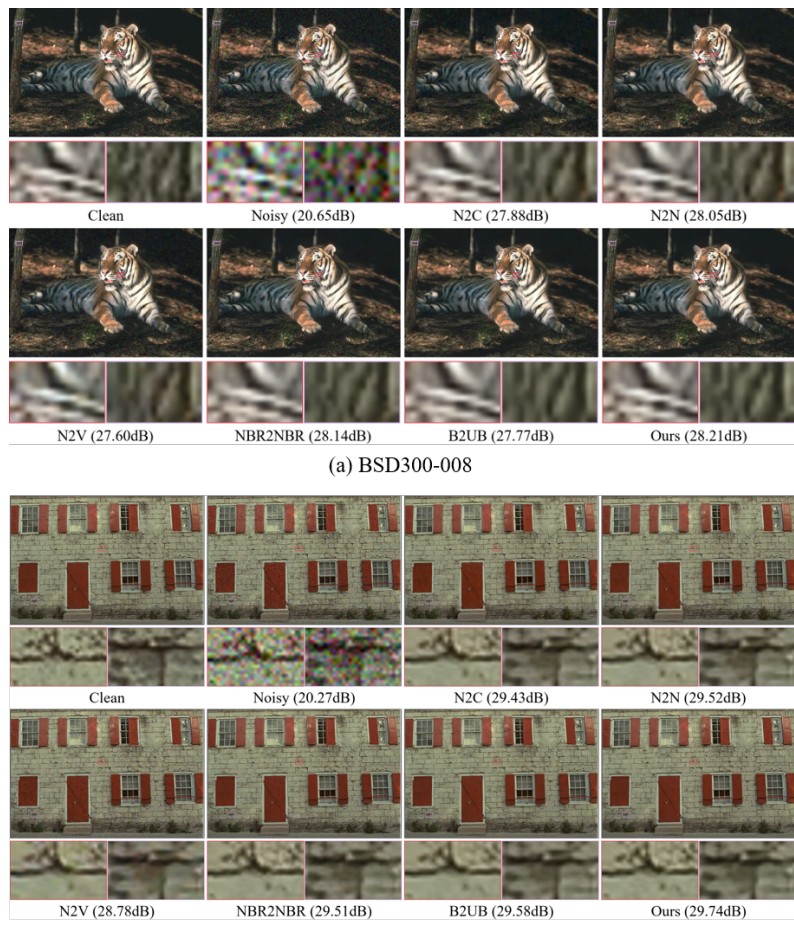

Figure 2. Visual comparison of denoising sRGB images in the setting of $\sigma = 25$.

### 4.2. Results for Real-World Denoising

In the real raw-RGB space, Table 2 shows the quality scores for the quantitative comparisons on the SIDD benchmark and SIDD validation. The SIDD website of the SIDD evaluates the quality scores for the SIDD Benchmark. Surprisingly, the proposed method outperforms the state of the art (NBR2NBR) by 0.28 and 0.23 dB for the benchmark and validation. It also outperforms the N2C and N2N by about 0.1 dB. It is worth noting that the unsupervised methods [30] and [15] relying on the model prior are significantly less effective when it comes to dealing with real noise, and we even surpass [15] by 4.05 and 4.09 dB for the benchmark and validation. Obviously, this type of model prior-based approach is not advisable. The raw-RGB denoising performance in the real world demonstrates that our method is able to simulate complex real noise distributions.

**Table 2.** Quantitative denoising results on SIDD benchmark and validation datasets in raw-RGB space.

| Method | Network | SIDD Benchmark | SIDD Validation |
|:---:|:---:|:---:|:---:|
| Baseline, N2C [21] | U-Net | 50.60/0.991 | 51.19/0.991 |
| Baseline, N2N [6] | U-Net | 50.62/0.991 | 51.21/0.991 |
| BM3D [29] | - | 48.60/0.986 | 48.92/0.986 |
| N2V [3] | U-Net | 48.01/0.983 | 48.55/0.984 |
| Laine19-mu [30] | U-Net | 49.82/0.989 | 50.44/0.990 |
| Laine19-pme [30] | U-Net | 42.17/0.935 | 42.87/0.939 |
| DBSN [31] | DBSN | 49.56/0.987 | 50.13/0.988 |
| NBR2NBR [14] | U-Net | 50.47/0.990 | 51.06/0.991 |
| R2R [15] | U-Net | 46.70/0.978 | 47.20/0.980 |
| Ours | U-Net | **50.75/0.991** | **51.29/0.991** |

Our method for denoising images in the real world is shown in Figure 3, which validates our conclusions. The denoising performance of our method is significantly better than that of the N2C and N2N, which require paired inputs, and slightly better than the R2R, which requires prior noise.

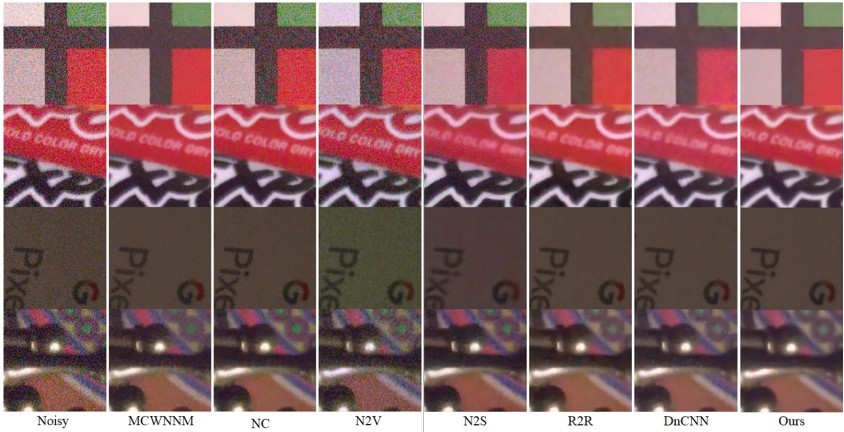

Noisy    MCWNNM    NC    N2V    N2S    R2R    DnCNN    Ours

**Figure 3.** Visual comparison of real-world denoising effects on SIDD dataset.

*4.3. Ablation Study*

This section conducts ablation studies on the pseudo supervised module. Table 3 lists the performance of different $\gamma$ values in unsupervised denoising, and the magnitude of the $\gamma$ values controls the strength of the pseudo supervised terms. When $\gamma = 0$, i.e., no pseudo supervised module is involved in the training, the denoising effect is poor. This is due to the fact that the model introduces extra noise, and the generated noise is slightly inaccurate. The pseudo supervised module improves the noise processing when $\gamma > 0$. We control $\gamma$ to increase gradually, and our method achieves the highest accuracy on the SIDD at $\gamma = 5$.

**Table 3.** Quantitative denoising results of different $\gamma$ on SIDD validation datasets.

| $\gamma$ | 0 | 2 | 5 | 10 | 15 | 30 |
|:---:|:---:|:---:|:---:|:---:|:---:|:---:|
| PSNR/SSIM | 49.41/0.989 | 51.23/0.991 | **51.29/0.991** | 51.23/0.991 | 51.20/0.991 | 51.06/0.991 |

In order to verify the effect of the generation noise module on the model, we denote the model without the generation noise module as GR2R/o and, conversely, as GR2R/w. As can be seen in Table 4, the denoising effect of GR2R/o is not as good as that of GR2R/w on the three different sets of network structures. Moreover, in order to verify the effect of the network structure on the denoising performance of the model, we compare the U-Net, ResNet and DensNet with three different network structures. The experimental

results are shown in Table 4, and the results show that the U-Net we use results in better results.

**Table 4.** Quantitative denoising results with or without generative noise module on SIDD validation datasets.

| Network | GR2R/o | GR2R/w |
| --- | --- | --- |
| ResNet | 49.56/0.935 | 50.84/0.973 |
| DensNet | 49.63/0.956 | 50.89/0.968 |
| U-Net | 50.15/0.947 | **51.29/0.991** |

### 5. Discussion

In this section, we summarize the results obtained and the findings of the overall paper.

a. The approach without prior noise is better for real-world denoising. According to the analysis of the results in Sections 4.1 and 4.2, our method is close to but not as good as the method showing noise modeling (Laine19-pme [30]) in terms of the denoising effects on a single distribution of noise, such as Gaussian noise and Poisson noise. However, the denoising effect in the real world is significantly improved. This is because we generate random dynamic noise during the training process, which more closely approximates real-world noise.

b. The pseudo supervision has a good suppression effect on noise. The hyperparameters are adjusted to observe the effect of the model with and without pseudo supervision loss and the different coefficients of the pseudo supervision loss. We find that the performance of the model without pseudo supervisory loss is severely degraded. In addition, different coefficients have different effects on the model, and the results show that the model performs best when the coefficient is five.

### 6. Conclusions

We propose the generative Recorrupted2Recorrupted, a novel unsupervised denoising framework, which achieves an excellent denoising performance without prior noise, and it surpasses methods that require prior noise. The proposed method generates random dynamic noise in the process of training the neural network so as to solve the problem of requiring a noise model prior to unsupervised denoising. In addition, the pseudo supervised module improves the performance of unsupervised denoising. Lastly, extensive experiments demonstrated the superiority of our approach compared to other methods.

**Author Contributions:** Conceptualization, Y.L.; methodology, Y.L.; software, Y.L.; validation, Y.L. and B.W.; formal analysis, Y.L.; investigation, B.W.; resources, D.S.; data curation, D.S.; writing—original draft preparation, B.W; writing—review and editing, D.S.; visualization, D.S.; supervision, D.S.; project administration, X.C.; funding acquisition, D.S. All authors have read and agreed to the published version of the manuscript.

**Funding:** This work is supported by the Ministry of Science and Technology China (MOST) Major Program on New Generation of Artificial Intelligence 2030 No. 2018AAA0102200. It is also supported by the Natural Science Foundation China (NSFC) Major Project No. 61827814 and the Shenzhen Science and Technology Innovation Commission (SZSTI) Project No. JCYJ20190808153619413. The experiments in this work were conducted at the National Engineering Laboratory for Big Data System Computing Technology, China.

**Data Availability Statement:** Not applicable.

**Conflicts of Interest:** The authors declare no conflicts of interest.

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
