# Peer review of "Generative Recorrupted-to-Recorrupted: An Unsupervised Image Denoising Network for Arbitrary Noise Distribution"

_remotesensing, doi:10.3390/rs15020364_

Round 1

Reviewer 1 Report

The article lacks a slight explanation of the research background, and suggests that the research results in this field in the past three years should be elaborated.

Author Response

Thank you for your comments, we have added a summary of the last three years of denoising technology in the first paragraph of Section 1.

Reviewer 2 Report

This paper proposed a denoising solution based on an unsupervised strategy. Although this topic is interesting, the presentation is poor and some necessary explanations and analyses are missing. Besides, there need to be more imaging results. So, I don't recommend its publication with the current version.

  1. As the authors proposed, they only considered the Gaussian noise. Yet, the noise types in real-world RGB-images may be others or not a single distribution. What happens if the noise types are not Gaussian?
  2. In figure 1, "generative noise" or "generate noise"? In the explanation of figure 1, "the generate noise" or "the generative noise"?
  3. From Table 1, it seems that Laine19-pme has competitive results. So, what are the advantages (in other terms) of the proposed methods, compared with Laine19-pme? Something is explained in Section. 4.2, but more explanations may be helpful.
  4. Figure 2 shows no significant difference in the visual comparison results. Please add text descriptions to explain the difference. And please show the visual results for real-world denoising.
  5. Section 5 is too thin! Please rewrite it and propose a more useful discussion.
  6. The ablation study is too thin. Please add more discussion. e.g., the impact of the Pseudo Supervised module on the network. Please add to the ablation study.

Author Response

Detailed point-by-point response to comments by Reviewer #2:

  1. This paper proposed a denoising solution based on an unsupervised strategy. Although this topic is interesting, the presentation is poor and some necessary explanations and analyses are missing. Besides, there need to be more imaging results. So, I don't recommend its publication with the current version.

A:   Thank you for the kind words.

  1. As the authors proposed, they only considered the Gaussian noise. Yet, the noise types in real-world RGB-images may be others or not a single distribution. What happens if the noise types are not Gaussian?

A:   We have added Poisson noise for comparison experiments in Section 4.1. From the Table 1, the results show that our method performs better than other unsupervised denoising without noise prior, and even surpasses some methods with noise prior.  

  1. In figure 1, "generative noise" or "generate noise"? In the explanation of figure 1, "the generate noise" or "the generative noise"?

A:        generative noise.

  1. From Table 1, it seems that Laine19-pme has competitive results. So, what are the advantages (in other terms) of the proposed methods, compared with Laine19-pme? Something is explained in Section. 4.2, but more explanations may be helpful.

A:        We give more explanation in Section4.1. The laine19-pme is a method that requires a noise prior. However, explicit noise modeling means strong prior, leading to poor performance on real data. And our method is to iteratively keep learning to learn the generated noise closer to the real data rather than some single distribution of noise. Therefore, for Gaussian or Poisson noise, our denoising effect is slightly insufficient.

  1. Figure 2 shows no significant difference in the visual comparison results. Please add text descriptions to explain the difference. And please show the visual results for real-world denoising.

A:        We give more explanation for the visual comparison results in Section4.1. Specifically, N2C, N2N and N2V are visually smoother than clean images, while NBR2NBR and B2UB are more detailed with our method of denoising, i.e., natural images are more detailed. Our method, however, works better if oversize to see more details than NBR2NBR and B2UB. And then, the Figure 3 show the visual results for real-world denoising in Section 4.2.

  1. Section 5 is too thin! Please rewrite it and propose a more useful discussion.

A:        We give a more useful discussion of the results of the experiments in Section 5.

  1. The ablation study is too thin. Please add more discussion. e.g., the impact of the Pseudo Supervised module on the network. Please add to the ablation study.

A:   We provide more discussion about pseudo-supervised ablation experiments in Section 4.3. This subsection conducts ablation studies on the pseudo supervised module. Table 3 lists the performance of different ? values in unsupervised denoising, and the magnitude of ? values controls the strength of the pseudo-supervised terms. When ? = 0, i.e., no pseudo-supervised module is involved in the training, the denoising effect is poor. This is due to the fact that the model introduces extra noise and the generated noise is slightly inaccurate. The pseudo-supervised module improves noise processing when ? > 0. We control ? to increase gradually, and our method achieves the highest accuracy on SIDD at ? = 5.

Reviewer 3 Report

The authors propose a denoising method which can be trained without clean images as training labels. The proposed method can be used for practical aims and experimental results verify that the proposed method is valid. However, the writing of the paper needs a lot of polishing to make it more readable for readers.

1. The main idea seems to be the adding of the "Generate Noise Module" and the "Pseudo Supervised" module. It would be good to show some results without the Recorrupted2Recorrupted Module and only the "Generate Noise Module" and the "Pseudo Supervised" module to show the functions of these modules. 

2. Some minor comments are as follows:

3 The premise of this method is effective 

is that there is certain data support, but in practice, it is particularly difficult to obtain labels in the training data. --> revise this statement.

53 denoising mode ---> denoising model

60 At this point, one epoch of the entire training ends. ---> revise this statement

61 we will gradually generate more realistic noise model and perfect denoiser. --> revise this statement

65 followed by the experiments in Section 4, in Section, --> revise the statement

143 Next, going a step further, how to achieve unsupervised denoising without noise prior?

--> revise the statement

161 where fθ a denoising network with arbitrary network design, ---> where fθ is a denoising network with arbitrary network design

182 We also GR2R with a traditional approach ---> revise the statement

Equation (2) does not match the statement above it, i.e., the statement which says "So the gradient generated by the network for one corrupted target is incorrect, but the gradient corresponding to the average of all corrupted images is correct, which can be expressed by the following equation:"

I cannot see why equation (2) represents this statement.

Which line refers to equation (7) on page 5?

The numbering of (6) and (7) are confusing.

Author Response

Detailed point-by-point response to comments by Reviewer #3:

  1. The authors propose a denoising method which can be trained without clean images as training labels. The proposed method can be used for practical aims and experimental results verify that the proposed method is valid. However, the writing of the paper needs a lot of polishing to make it more readable for readers.

A:   Thank you for your recognition of our work.

  1. The main idea seems to be the adding of the "Generate Noise Module" and the "Pseudo Supervised" module. It would be good to show some results without the Recorrupted2Recorrupted Module and only the "Generate Noise Module" and the "Pseudo Supervised" module to show the functions of these modules.

A:        Indeed, the core innovation of our approach is the "Generate Noise Module" and the "Pseudo Supervised" module, however, the Recorrupted2Recorrupted Module is the basis of our unsupervised approach, without the unsupervised module, we would need pairs of inputs or noise priors, which our approach discards. Therefore, we do not use only the "Generate Noise Module" and the "Pseudo Supervised" module.

  1. Some minor comments are as follows: …

A:        All suggestions have been accepted and the statement has been revised.

  1. Equation (2) does not match the statement above it, i.e., the statement which says "So the gradient generated by the network for one corrupted target is incorrect, but the gradient corresponding to the average of all corrupted images is correct, which can be expressed by the following equation:" I cannot see why equation (2) represents this statement.

A:        We corrected the description and equation (2) in Section 2.1. Since N2N only requires that the ground truth be "clean" at some statistical values, and does not require that every ground truth be "clean", the optimal solution of this loss function is obtained at the arithmetic mean of the measured values (expectation) of the measurements. Therefore, we add an expectation value to equation (2). ( please refer to International Conference on Machine Learning, 2018, pp. 2965–2974.)

  1. Which line refers to equation (7) on page 5? The numbering of (6) and (7) are confusing.

A:        We modified the formulation of equation (6) and (7) in Section 3.1. The details are as follows:

Round 2

Reviewer 2 Report

Although most of my concerns were well addressed, some replies must be improved before recommending the work's publication.

  1. The authors added the results of Poisson noise, but the related words in line 185 are poor and confusing, e.g., what is \lamda \in [5 50]? Besides, according to Table 1, the comparison between Laine19 and yours should be explained more clearly; i.e., the reader needs to see more explanation.
  2. Please introduce your real-world data in Fig. 3 more clearly. e.g., the data's sources, characteristics, noise type, etc.
  3. The section of the ablation study is still thin. Can you give more results, except the \sigma, such as the network structure, other network parameters, and noise characteristics?

Author Response

1. The authors added the results of Poisson noise, but the related words in line 185 are poor and confusing, e.g., what is \lamda \in [5 50]? Besides, according to Table 1, the comparison between Laine19 and yours should be explained more clearly;  i.e., the reader needs to see more explanation.

A:        We are very sorry for the confusion. We have explained \lamda \in [5,50] in more detail. In the experimental setup, we use 4 different noise settings. And \lamda \in [5,50] means that the Gaussian noise λ takes random values within a variation range [5,50]. Moreover, in Section 4.1, we added some explanations for the results of the comparison experiments. In addition to the reason that Laine19 is shown to model noise, compared to Laine19, we use mask to train the noise model, which may ignore the central pixel, resulting in only a few output pixels that can contribute to the loss function.

2. Please introduce your real-world data in Fig. 3 more clearly. e.g., the data's sources, characteristics, noise type, etc.

A:        The real dataset in Figure 3 is from SIDD, and we add more information about the SIDD real dataset in Section 4. Specifically, the Smartphone Image Denoising Dataset (SIDD) dataset collected by five smartphone cameras in 10 scenes under different lighting conditions for real-world denoising in raw-RGB space, which has about 30,000 noisy images and 200 scene instances, of which 160 scene instances are used as training set and 40 scene instances are used as test set.

3.  The section of the ablation study is still thin. Can you give more results, except the \sigma, such as the network structure, other network parameters, and noise characteristics?

A:        We add more ablation experiments in Section 4.3, including the effect of using different network structures and whether or not to use the generating noise module on the model. For the specific experimental results, please see Table 4.

Reviewer 3 Report

The authors have revised the paper according to my former comments. However, still the paper needs some editing of the english language. I suggest that the authors undergo some english editting service before publishing their paper. 

Author Response

1.  The authors have revised the paper according to my former comments. However, still the paper needs some editing of the english language. I suggest that the authors undergo some english editting service before publishing their paper.

A:        Thank you for your recognition of my work, I have edited the whole paper in English through the professional English editor of MDPI, and this ID is english-edited-57082.